# A Note on Cherry-Picking in Meta-Analyses

**DOI:** 10.3390/e25040691

**Published:** 2023-04-19

**Authors:** Daisuke Yoneoka, Bastian Rieck

**Affiliations:** 1Center for Surveillance, Immunization, and Epidemiologic Research, National Institute of Infectious Diseases, Tokyo 162-8640, Japan; 2Institute of AI for Health, Helmholtz Munich, Technical University of Munich, 80333 Munich, Germany

**Keywords:** meta-analysis, cherry-picking studies, selection bias, adversarial meta-analysis, inclusion/exclusion criteria

## Abstract

We study selection bias in meta-analyses by assuming the presence of researchers (meta-analysts) who intentionally or unintentionally cherry-pick a subset of studies by defining arbitrary inclusion and/or exclusion criteria that will lead to their desired results. When the number of studies is sufficiently large, we theoretically show that a meta-analysts might falsely obtain (non)significant overall treatment effects, regardless of the actual effectiveness of a treatment. We analyze all theoretical findings based on extensive simulation experiments and practical clinical examples. Numerical evaluations demonstrate that the standard method for meta-analyses has the potential to be cherry-picked.

## 1. Introduction

Meta-analysis is a methodology for evaluating the overall treatment effect by integrating the results of past clinical trials and is widely recognized as one of the research methods that underlie “Evidence Based Medicine” [1,2]. Generally, the methodology involves the integration of summary statistics, such as odds ratios or hazard ratios reported in published papers, by using appropriate statistical methods to estimate the average treatment effect [1,2,3]. In a meta-analysis, various biases that could affect the validity of the synthesized results have been widely studied, for example, (1) *publication bias*, whereby positive results are more likely than negative or null results to be published [4]; (2) *language bias*, whereby non-English studies tend to be excluded from meta-analyses [5]; (3) *time-lag bias*, whereby positive results tend to have longer time differences from trial completion to publication than negative or null results [6]; (4) *reporting bias*, whereby studies selectively report outcomes favoring their hypothesis [1]; (5) *outlier bias*, whereby a single or a few studies disproportionately influence the overall results of a meta-analysis [7]; (6) *categorization bias*, whereby studies use different categorization or stratification schemes to achieve the same outcome [8]; and (7) *covariate set bias*, whereby studies use different covariate sets in the regression model that share the same regression task across the studies [9]. In this study, we aim to focus on a new source of bias, the “cherry-picking” bias, in meta-analyses.

The simplest setup of a meta-analysis is to assume that there are *K* independent studies, each yielding an estimate $y_{i}$ (i=1,⋯,K) of an underlying treatment effect parameter θ. The standard fixed-effect model is defined as
(1)$$\begin{array}{c} y_{i}∼N\left(\theta ,\sigma_{i}^{2}\right), \end{array}$$
where $\sigma_{i}^{2}$ is the reported (known) within-study variance of the *i*th study. Under this fixed-effect model, the maximum likelihood estimate of θ is defined by the weighted average
(2)$$\begin{array}{c} \hat{\theta }=\frac{\sum_{i=1}^{K}w_{i}y_{i}}{\sum_{i=1}^{K}w_{i}}, \end{array}$$
where the *i*th study is assigned the weight $w_{i}=1/\sigma_{i}^{2}$. The corresponding standard normal test statistic is
$$\begin{array}{c} T\left(\theta \right)=\sqrt{\sum_{i=1}^{K}w_{i}}\left(\hat{\theta }-\theta \right), \end{array}$$
and the resulting confidence interval (CI) is
$$\begin{array}{c} \left\{\theta :|T\left(\theta \right)\right|\leq z_{\alpha }\}=\left[\hat{\theta }-z_{\alpha }{\left(\sum w_{i}\right)}^{-1/2},\hat{\theta }+z_{\alpha }{\left(\sum w_{i}\right)}^{-1/2}\right], \end{array}$$
where $z_{\alpha }=\Phi^{-1}\left(1-\alpha /2\right)$ is the standard normal percentage point for the coverage of 1−α, and Φ is the cumulative distribution function of the standard normal distribution [10,11,12,13]. In practice, researchers are often interested in a hypothesis regarding whether a given treatment has no effect ($H_{0}:\theta =0$) or is beneficial ($H_{1}:\theta >0$). The one-sided *p*-value of the *i*th study is defined as
(3)$$\begin{array}{c} p_{i}=\Phi \left(-\sqrt{w_{i}}y_{i}\right). \end{array}$$Similarly, the *p*-value of Equation (Equation 2) is defined as
(4)$$\begin{array}{c} p_{meta}=\Phi \left(-\hat{\theta }\sqrt{\sum_{i=1}^{K}w_{i}}\right). \end{array}$$Equation (Equation 1) is based on the fixed-effect assumption that each study shares the same underlying effect θ. When heterogeneity between included studies is suspected, the random-effect model is fitted as
(5)$$\begin{array}{c} y_{i}∼N\left(\theta ,\sigma_{i}^{2}+\tau^{2}\right), \end{array}$$
where $\tau^{2}$ is the between-study variance, which can be estimated from the data using standard methods, such as the method proposed in DerSimonian and Laird [3]. The same reasoning can be applied by replacing the weights $w_{i}$ in Equation (Equation 2) with
(6)$$\begin{array}{c} w_{i}=\frac{1}{\sigma_{i}^{2}+\tau^{2}}. \end{array}$$Refer to the studies by [10,11,12] and Cooper et al. [13] for a detailed discussion of the various methods used for meta-analysis.

One of the most important stages of a meta-analysis is the specification of the inclusion and/or exclusion criteria, because the selection of studies for a literature review is known to influence the conclusions. One must carefully consider which studies to include or exclude from the review to obtain unbiased and fair conclusions. However, in reality, a significant number of meta-analyses are published without a protocol to define the inclusion and exclusion criteria before conducting the meta-analysis and systematic review. Furthermore, it is not common for papers to follow procedures such as stating inclusion and exclusion criteria in advance and adhering to them. For example, Page et al., (2016) examined the reporting completeness of Biomedical Research meta-analyses and found that only 16% of the included reviews had a publicly accessible protocol published before the review was conducted [14]. In addition, Tawfic et al., (2020) found that only 37.4% of researchers who are trying to conduct a meta-analysis agree that protocol registration prior to the main analysis should be mandatory [15].

Given a set of included studies, the conclusions obtained from the results of meta-analyses are frequently based on statistical tests and their associated *p*-values in practice. Ideally, a statistical test with a type 1 error rate of α should be used to control the ratio of false findings at a ratio of (less than) α. However, inclusion and/or exclusion criteria can be misused by (sometimes malicious) meta-analysts (i.e., the authors of a meta-analysis who intentionally or unintentionally report false (non)significant overall effects, regardless of the actual treatment effect) to pick a subset of all studies that changes the result and sometimes leads to their desired conclusion. This practice is also known as cherry-picking, and it means that the resulting *p*-value no longer controls the ratio of false findings. Figures in Section 4 show practical examples. Reviewer selection bias is also known in the field of meta-analysis as the situation where reviewers (un)intentionally seek only a subset of existing studies that satisfy certain criteria, so the chosen subset does not reflect all available evidence [16]. The degree of bias in a synthesized result can depend on a selector’s prior knowledge, research field, existing collaborators, and opinion regarding the research question of interest [17]. Other similar biases related to inclusion and/or exclusion criteria include the English language bias (whereby non-English studies are more likely to be excluded), the data availability bias (whereby only studies with individual patient data are included), and the database bias (whereby only studies published in journals indexed in popular databases such as Embase or Medline are included); see [17,18,19] for an overview of this topic. For instance, Ahmed et al. [17] investigated 31 meta-analyses and found that 29% of them suffered from a significant selection bias based on the use of selective or nonsystematic approaches for the identification of relevant studies. They concluded that biased synthesized results can lead to incorrect decisions by medical practitioners, which can harm patients because inefficient or ineffective treatments may be chosen. Such results can also mislead future research efforts [20]. However, although the selection bias has a similar impact on synthesized results to the publication bias, which has been widely studied in the field of meta-analysis, no attempts have been made to examine the selection bias from a statistical perspective. In this study, we demonstrate that it is possible to modify the results of a meta-analysis by changing the inclusion and/or exclusion criteria to select an arbitrary subset of studies, so that they support a biased conclusion, such as (i) the treatment of interest having a significant effect, despite there being no actual effect or (ii) the treatment having a nonsignificant effect, despite the presence of an actual effect. The reliability of a meta-analysis is decreased in the presence of such a selection bias. The goal of this study is to identify the possibility of cherry-picking.

The remainder of the article is organized as follows: In Section 2, we show theoretical guarantees on the chance of cherry-picking by meta-analysts who intentionally or unintentionally select the subset of studies. To demonstrate that conventional meta-analysis procedures have a significant cherry-picking effect, the results of extensive simulation studies are presented in Section 3, and two clinical datasets are examined in Section 4. Lastly, Section 5 presents a discussion and our conclusions.

## 2. Methods

We consider the simple fixed-effect meta-analysis settings defined in Equation (Equation 1). An extension for a random-effect model is described in Section 2.2 and later in the discussion section. We assume that there are *K* studies $D_{K}=\left\{1,2,\cdots ,K\right\}$ collected via data extraction from several databases such as PubMed, Medline, and Embase. Each study is supposed to report an estimate $y_{i}$ and corresponding variance $\sigma_{i}^{2}$ (or $w_{i}$, equivalently). Meta-analysts determine the inclusion and/or exclusion criteria to select a subset of *S* studies from all *K* studies found in the databases. This subset is denoted as $D_{S}=\left\{1,2,\cdots ,S\right\}\subseteq D_{K}$. Therefore, $D_{S}$ may suffer from a selection bias. In this study, we assume that meta-analysts (intentionally or unintentionally) select studies $D_{S}$ to (i) overstate the effect of the treatment of interest (Case 1), despite the treatment having no actual effect (i.e., θ=0), or (ii) understate the effect of the treatment (Case 2), despite the treatment having an actual effect (i.e., θ>0). Furthermore, we assume that meta-analysts use a statistical testing framework by defining the null and alternative hypotheses as $H_{0}:\theta =0$ and $H_{1}:\theta >0$, respectively. The null hypothesis $H_{0}$ states that the treatment has no effect, while the alternative hypothesis $H_{1}$ states that the treatment has a significant effect. Statistical significance at a level of α∈(0,1) for the dataset $D_{S}$ is defined as
(7)$$p_{meta}\left(D_{S}\right)=\Phi \left(-\sqrt{\sum_{i\in D_{S}}w_{i}}\hat{\theta }\right)\leq \alpha ,$$
where $\Phi \left(x\right)=\int_{-\infty }^{x}\phi \left(t\right)dt$, $\phi \left(x\right)=\left(1/\sqrt{2\pi }\right)\exp\left(-x^{2}/2\right)$, and $w_{i}=1/\sigma_{i}^{2}$ or Equation (Equation 6) is used in the fixed- and random-effect models, respectively. The extension to two-sided tests is easy and is discussed later in Section 5.

### 2.1. Chance of Cherry-Picking in a Meta-Analysis

This section describes how the standard hypothesis testing procedure is no longer robust against selection bias due to the cherry-picking of studies using biased inclusion and/or exclusion criteria. We used similar techniques to those employed by Komiyama and Maehara [21] in the following derivation.

Theorem 1 guarantees that, under certain mild conditions, it is possible for meta-analysts to have sufficient statistical power to falsely conclude that a significant effect of the treatment of interest (Case 1) exists, even if the treatment has no actual effect. This is achieved by cherry-picking the subset $D_{S}$ that provides the top-*S* smallest *p*-values.

**Theorem** **1.**
*For any α∈(0,1/2), δ∈(0,1), and ϵ∈(0,1/3), if S/K≤ϵ and*

$$S\geq \max\left(\eta {\left\{\frac{\Phi^{-1}\left(\alpha \right)}{\Phi^{-1}\left(\frac{1}{2}-\frac{\epsilon }{2}\right)}\right\}}^{2},\epsilon \left\{\frac{\log\left(\frac{1}{\delta }\right)}{2{\left(\frac{1}{2}-\frac{3\epsilon }{2}\right)}^{2}}-2\right\}\right),$$

*with $\eta =\frac{w_{max}}{w_{min}}$, $w_{max}=\max_{i\in D_{S}}w_{i}$ and $w_{min}=\min_{i\in D_{S}}w_{i}$, then meta-analysts can select $D_{S}$ such that $p_{meta}\left(D_{S}\right)\leq \alpha$ with a probability of at least 1−δ.*


Similarly, Theorem 2 guarantees that under certain conditions, it is also possible to falsely conclude that the treatment has an insignificant effect (Case 2), even if the treatment has an actual effect. This is achieved by cherry-picking the subset $D_{S}$ that provides the top-*S* largest *p*-values.

**Theorem** **2.**
*For any α∈(0,1/2), δ∈(0,1), and ϵ∈(0,1), if 1−ϵ≤S/K≤1 and*

$$\begin{array}{c} \eta {\left\{\frac{\Phi^{-1}\left(\alpha \right)}{\Phi^{-1}\left(1-\frac{\epsilon }{2}\right)}\right\}}^{2}\leq S<\left(1-\epsilon \right)\left\{\frac{\log\left(\frac{1}{\delta }\right)}{2{\left(1-\epsilon /2\right)}^{2}}-2\right\}, \end{array}$$

*the meta-analysts can select $D_{S}$ such that $p_{meta}\left(D_{S}\right)\geq \alpha$ with a probability of at least δ.*


Together, these theorems imply that meta-analysts have a chance to change the results of meta-analysis, regardless of real treatment effects, by cherry-picking an appropriate value for *S*. When *S* satisfies the conditions outlined in the theorems, readers or inspectors of the meta-analysis results can claim that the possibility of cherry-picking exists. In addition, now, we have assumed that meta-analysts cherry-pick the subset of studies $D_{S}$ yielding the “top-*S*” (largest/smallest) *p*-values, which sometimes seems an unrealistic assumption because the actual meta-analysts might try to cherry-pick the subset of studies in a more arbitrary manner. However, it is noteworthy that even if meta-analysts cherry-pick an arbitrary subset of all studies such as the subset of studies with moderate *p*-values, these theorems are still valid because the current assumption of $D_{S}$ is the most aggressive and worst setting, i.e., we assume that $D_{S}$ provides the minimum/maximum *p*-value in the proof. Thus, the theorem still holds even under the more relaxed assumption of cherry-picking moderate *p*-values. The proofs for the theorems can be found in Appendix A, Appendix B and Appendix C.

### 2.2. Extension to a Random-Effect Model

In the above section, we tentatively assumed that $w_{i}$ was known, which corresponds to a fixed-effect model in a meta-analysis. However, we can also consider cases in which $\tau^{2}$ in Equation (Equation 6) is estimated. In other words, we can estimate the between-study variance using the random-effect model. In practice, $\tau^{2}$ is estimated from the data, frequently by using the method proposed by [3]. Given $D_{S}$, the DerSimonian–Laird estimate of $\tau^{2}$ is defined as
(8)$$\max\left[0,\frac{\sum_{i\in D_{S}}w_{i}{\left(y_{i}-y_{0}\right)}^{2}-S+1}{\sum_{i\in D_{S}}w_{i}-\sum_{i\in D_{S}}w_{i}^{2}/\sum_{i\in D_{S}}w_{i}}\right],$$
where $y_{0}=\sum_{i\in D_{S}}w_{i0}y_{i}/\sum_{i\in D_{S}}w_{i0}$ and $w_{i0}=1/\sigma_{i}^{2}$. Theorems 1 and 2 are nontrivial because this estimate depends on the choice of $D_{S}$, and the selection of the top-*S* largest test statistics of $\sqrt{w_{i}}y_{i}$ depends on the estimate. These factors eliminate the simplicity of Theorems 1 and 2 and require a more sophisticated analysis. One possible approach is that, instead of using Equation (Equation 8), we replace $D_{S}$ and *S* in Equation (Equation 8) with $D_{K}$ and *K*, respectively. This corresponds to the situation where once $\tau^{2}$ is estimated, it is regarded as a fixed constant in the model and the same discussion is applied with Theorems 1 and 2. The results of the random-effect models are examined in the simulation and application sections. In addition, in our future work, we plan to extend our results to cover cases in which $\tau^{2}$ depends on the choice of $D_{S}$.

## 3. Simulation Experiments

### 3.1. Simulation Settings

In this section, we describe Monte Carlo simulations that were implemented to demonstrate how sensitive the standard hypothesis test for meta-analyses is to the cherry-picking of studies, allowing meta-analysts to derive biased conclusions.

We considered both Case 1, where meta-analysts try to overstate the effectiveness of a treatment, despite there being no actual effect and Case 2, where meta-analysts try to understate the effectiveness of a treatment, despite there being an actual effect. The tunable parameters for the simulation scenarios were the number of cherry-picked studies S=2,⋯,30, the proportion of cherry-picked studies among all studies S/K∈{1/3,1/5,1/10}, the true treatment effect θ∈{0,0.5,1.0}, and the between-study variance $\tau^{2}\in \left\{0,0.01,0.10,0.50,0.70\right\}$, where $\tau^{2}=0$ corresponds to the fixed-effect model and $\tau^{2}>0$ corresponds to the random-effect model. Additionally, θ=0 corresponds to Case 1 and θ>0 corresponds to Case 2. Following the approach described by Brockwell et al., (2001) [22], we simulated *K* independent studies. Each study has $y_{i}$ and $\sigma_{i}^{2}$, where $y_{i}$ and $\sigma_{i}^{2}$ are assumed to follow
$$\begin{array}{c} y_{i}|\sigma_{i}∼N\left(\theta ,\sigma_{i}^{2}+\tau^{2}\right),\;\;\sigma_{i}^{2}∼0.25\chi_{1}^{2}. \end{array}$$The variances $\sigma_{i}^{2}$ were assumed to follow a $\chi_{1}^{2}$ distribution, multiplied by 0.25 and truncated to an interval of (0.009,0.600), resulting in a mean within-study variance estimate of 0.17 [22]. Because $\tau^{2}$ was varied from 0 to 0.70, the heterogeneity measure $I^{2}$ moved from 0% (no heterogeneity) to 80% (considerable heterogeneity). Throughout our simulations, we used α=0.05 as the type 1 error rate. Using these settings, we performed 1000 Monte Carlo simulations. In addition, values of S>30, were examined, but they did not yield any notably different results. Therefore, we excluded the results for these settings.

### 3.2. Simulation Results

The simulation results revealed that when meta-analysts try to cherry-pick studies to change (or manipulate) pooled estimates and obtain their preferred conclusions, they have a chance of making it work in practice. Figure 1 and Figure 2 present the simulation results for Cases 1 and 2, respectively. They present the proportions of false conclusions (i.e., the proportion of 1000 iterations that succeeded in “flipping” the conclusion from significant to nonsignificant).

Figure 1 presents the results for Case 1, where the true treatment effect is θ=0. This indicates that, when using the random-effect model ($\tau^{2}=0.1,0.5$), the proportion of false conclusions increases as *S* increases or S/K decreases in the standard hypothesis testing framework. In particular, when $\tau^{2}$ is large, it is possible for meta-analysts to almost always cherry-pick studies to falsely conclude a significant treatment effect, despite there being no actual effect. When using the fixed-effect model ($\tau^{2}=0$), a similar tendency was observed: the proportion of false conclusions increased as *S* increased or S/K decreased. Figure 2 also presents the results for Case 2 where the true treatment effects are θ=0.5 and 1.0, respectively. This shows that meta-analysts still have a chance of cherry-picking studies to falsely conclude treatment insignificance, despite there being an actual effect. However, it shows different trends from Case 1: the proportion of false conclusions sometimes decreases as *S* increases. Especially when τ is small (τ=0 or 0.01) and θ is large, the standard hypothesis testing works well as *S* increases.

## 4. Medical Application Studies

This section shows how we can cherry-pick studies from two medical datasets. We emphasize that the original and subsequent analyses in the referenced articles were not cherry-picked. However, since cherry-picking is not reported in practice by definition, it is impossible to obtain real cherry-picked examples. Therefore, we made artificially cherry-picked situations from these real-world datasets, which are described in the following subsections.

### 4.1. Case 1 Example: Clinical Trials on the Effectiveness of Magnesium for Reducing the Mortality of Acute Myocardial Infarction Patients

We considered the results of randomized clinical trials (RCTs) that tested the effectiveness of intravenous magnesium for reducing the mortality following acute myocardial infarction (AMI). Because magnesium has been shown to protect ischemic myocytes from calcium overload, it is of significant interest to examine how magnesium can affect the mortality of ischemic heart disease patients. Teo et al. [23] conducted a meta-analysis using a fixed-effect approach based on seven studies (studies 1–7 in Figure 3), suggesting that magnesium has a significant effect on reducing the mortality of AMI patients. However, the results of a large trial (ISIS-4) indicated contradicting results, namely, that magnesium has no significant effect on reducing mortality [24]. We considered 16 studies that reported their summary statistics and the estimated odds ratio. They were extracted from Eggar and Smith [25], including ISIS-4 and the seven studies in Teo et al. [23]. Figure 3 presents the synthesized results when using the fixed-effect model presented by Teo et al. [23]. It is shown that magnesium has no significant effect on reducing mortality when considering all 16 studies; we obtained an estimated odds ratio (OR) (95% CI) of 0.994 (0.937, 1.055) and a *p*-value of 0.579 based on the one-sided test defined in Equation (Equation 3). Therefore, when using the 16 studies, there is a chance for meta-analysts to cherry-pick studies to obtain a biased treatment effect of magnesium (corresponds to Case 1). For example, when using the same set of studies as those used in Teo et al. [23], it appears that magnesium has a significant effect on mortality reduction with an estimated OR (95% CI) is 2.224 (1.401, 3.531) and *p*-value <0.001. Therefore, relying on the results of Teo et al. [23] to conclude the effectiveness of magnesium (hypothetically) corresponds to Case 1.

### 4.2. Case 2 Example: Clinical Trials on the Effectiveness of St. John’s Wort for Treating Depression

We considered the results of nine RCTs on the effectiveness of extracts of Hypericum perforatum (St. John’s wort) for treating depression. Originally considered to be an effective treatment for depression, there have been mixed findings from several clinical trials comparing St. John’s wort to a placebo. Linde et al. [26], from which we borrowed data, assessed a number of patients categorized as “responders” based on the Hamilton Rating Scale for Depression (HRSD). Notably, 17 studies were dropped from the group of studies used by Linde et al. [26] because they used a different version of the HRSD for assessing the degree of depression.

Figure 4 presents the results for the random-effect model. It simulates how meta-analysts can cherry-pick studies by selecting another definition of the treatment response (Def. 1: HRSD score reduction of at least 50% compared to baseline or HRSD score after therapy <10; Def. 2: HRSD reduction of at least 50% compared to baseline) to conclude the insignificant effectiveness of St. John’s wort for depression, regardless of the actual effect (i.e., this case study corresponds to Case 2): St. John’s wort provides a reduction in depression when considering all nine studies with an estimated OR (95% CI) of 1.467 (1.067, 2.016) and a *p*-value of 0.009. In contrast, when restricting our analysis to the subset of studies that applied only Def. 1 for the definition of a treatment response, we concluded that St. John’s wort has a nonsignificant effect in reducing depression with an estimated OR (95% CI) of 1.458 (0.753, 2.822) and a *p*-value of 0.132.

## 5. Discussion

The conclusions of any meta-analysis can be biased if meta-analysts intentionally or unintentionally cherry-pick a subset of all studies that lead to a desired favorable result. This is achieved by choosing beneficial inclusion and/or exclusion criteria. We theoretically assessed the conditions under which such cherry-picking is possible. To prevent cherry-picking in a meta-analysis, one solution is to mandate stricter adherence to Cochrane and other guidelines. This would require meta-analysts to register and publish their protocol before carrying out the primary meta-analysis. In addition, a more advanced mechanism would be necessary to verify the inclusion/exclusion criteria that were not initially included in the protocol but were subsequently added. The R code is provided in a GitHub repository (https://github.com/kingqwert/R/tree/master/metaCherry/, accessed on 2 March 2023) and will be hosted on the *R* CRAN repository (https://www.r-project.org/, accessed on 2 March 2023) in the near future, allowing others to apply our method easily.

Extensive Monte Carlo simulations were conducted to illustrate that the standard meta-analysis method could be subject to cherry-picking, leading to biased results. The chance of cherry-picking is remarkably high, especially when *S* is small. Furthermore, two real data analysis problems were simulated to provide new insights into the results of RCTs on the effectiveness of magnesium on AMI and St. John’s wort on depression. We demonstrated that it is easy to obtain favorable, i.e., biased, conclusions by cherry-picking studies based on biased inclusion and/or exclusion criteria. We encourage the re-evaluation of our approach using other datasets.

We demonstrated that meta-analysts can cherry-pick a subset of studies by modifying inclusion and/or exclusion criteria. However, this type of cherry-picking should not be taken too literally: the theorems presented in this study can be applied to any type of cherry-picking if information regarding *K*, *S*, and $w_{i}$ is available. In addition, we analyzed the case of cherry-picking from a ‘subset’ of studies (i.e., the case of S<K). It is trivial to extend this analysis to the case of K<S, where meta-analysts use unsuitable inclusion and/or exclusion criteria to increase the total number of studies to obtain a favorable conclusion. Similarly, although we focused on the case of one-sided right-tailed hypothesis testing in this study, it is simple to extend our results to (i) the one-sided left-tailed hypothesis case ($H_{0}=0$ and $H_{0}<0$) by using $p_{meta}=\Phi \left(\hat{\theta }\sqrt{\sum_{i\in D_{S}}w_{i}}\right)$, and (ii) the two-sided hypothesis case ($H_{0}=0$ and $H_{0}\neq 0$) by using $p_{meta}=2\Phi \left(-\hat{\theta }\sqrt{\sum_{i\in D_{S}}w_{i}}\right)$ instead of Equation (Equation 7). In addition, the assumption of cherry-picking the top-*S* results is sometimes unrealistic, and actual meta-analysts might try to cherry-pick the subset of studies in a more arbitrary manner. However, we note again that, as discussed in Section 2.1, the theorems are still valid, even if meta-analysts cherry-pick an arbitrary subset of all studies.

Similar to most published studies on meta-analyses, the within-study variance $\sigma_{i}^{2}$ in Equation (Equation 1) was assumed to be known, ignoring the fact that it must be estimated in practice. If the estimated $\sigma_{i}^{2}$ and $\tau^{2}$ values are used to define the *p*-value, it no longer follows a standard normal distribution under the null hypothesis [27,28,29], eliminating the simplicity of our theorems. In such cases, a more complicated asymptotic analysis would be required. Furthermore, there have been many previous attempts to formulate a “publication bias” using *p*-values [30,31]. It would be worthwhile to consider both selection and publication biases simultaneously by using the proposed framework for hypothesis testing and its associated *p*-value. However, further discussion about the conceptual difference between the publication bias and selection bias due to cherry-picking is required.

## Figures and Tables

**Figure 1 entropy-25-00691-f001:**
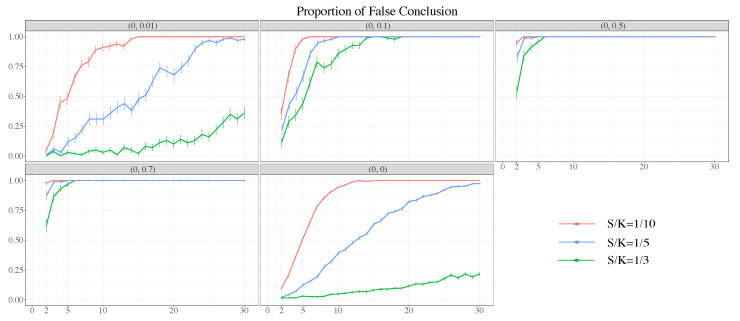
Simulation results for the standard hypotheses for Case 1, where meta-analysts overstate the effect of the treatment, regardless of there being no actual effect: Proportion of False Conclusions (i.e., type 1 error). (a,b) indicates θ=a and τ=b.

**Figure 2 entropy-25-00691-f002:**
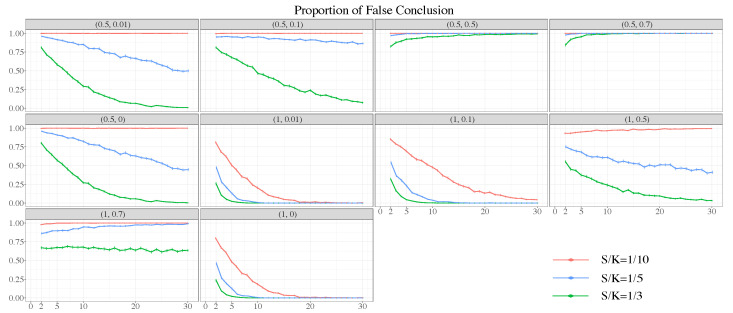
Simulation results for the standard hypotheses for Case 2, where meta-analysts understate the effect of the treatment, regardless of the actual effect: Proportion of False Conclusions (i.e., type 2 error). (a,b) indicates θ=a and τ=b.

**Figure 3 entropy-25-00691-f003:**
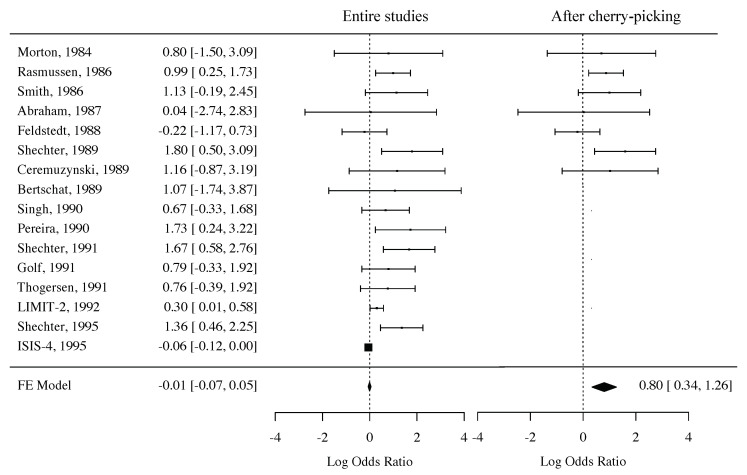
Meta-analysis of the results of 16 RCTs on the effectiveness of magnesium for reducing mortality following AMI.

**Figure 4 entropy-25-00691-f004:**
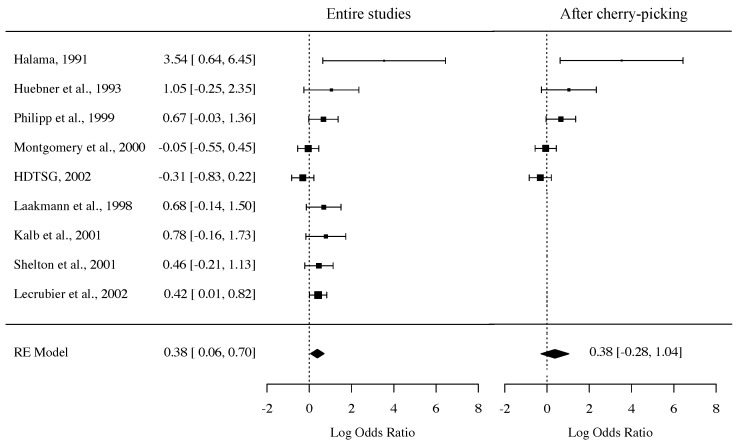
Meta-analysis of the results from nine RCTs on the effectiveness of St. John’s wort for treating depression.

## Data Availability

No new data were created or analyzed in this study. Data sharing is not applicable to this article.

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
