# Peer review of "A Note on Cherry-Picking in Meta-Analyses"

_entropy, 2023, doi:10.3390/e25040691_

Round 1

Reviewer 1 Report

First of all, I would like to thank the Editor of the journal "Entropy" for inviting me to review this impressive manuscript.

After a careful review of the manuscript (entropy-2290977) within the given time frame, I have reached the following opinions

# Overall opinion

I think it is an interesting manuscript with a detailed review of a topic that is always controversial in meta-analysis methodology, and it deserves to be published in this journal, both in terms of the completeness of the methodology and the novelty of the topic. I think it was a great learning experience for me as a reviewer to come across a manuscript that provides a detailed analysis of a topic that I usually only think about in my head while doing research.

# Major concerns

I found no major flaws in the logical structure of the study and the reliability of the conclusions.

# Minor concerns

1. it seems to me that the structure of the introduction needs to be slightly changed to improve its readability; for example, it would be a more reader-friendly manuscript if the challenges to be addressed in the current "meta-analysis" methodology were first defined narratively and then elaborated on using mathematical formulas.

2. I strongly agree with the author's views. However, I think readers would benefit from a more detailed discussion of how researchers conducting future meta-analyses should proceed. By adding one or more paragraphs to provide "guidelines for researchers", I think this article will be read and cited by many more researchers performing meta-analysis methods. I think it would be more welcomed by a wider audience if there was an example of how some of the Cochrane guidelines could be improved.

3. The authors state that "cherry-picking" is not formally defined, but it would be great if the authors could provide their screening guidelines for meta-analysis studies that have certain characteristics in results and methodology. This would be very helpful for the many health professionals who rely on meta-analyses for decision making.

4. I believe that some of the problems you raise in this manuscript could be overcome by a more aggressive use of Bayesian meta-analysis methods(NOT Bayesian network meta-analysis). If possible, I would be grateful if the authors could add their detailed views on this topic to the manuscript, as this would be another step forward in the conduct of meta-analysis.

I hope that my views will help to improve the manuscript and its publication.

Reviewer 2 Report

The author used mathematical formulae to show selection bias in meta analysis by arbitrary select top several results can make all meta-analysis show significance. This is a very straight and apparent conclusion. I think it is ok to support this argument using some mathematical formulae. The authors need to provide more details and literature on it.

Major Comments:

1. The authors only shows selection bias by intentionally select top several results in meta-analysis. However, cherry-picking is not allowed in meta-analysis. Thus, the authors need to verify why a standard meta-analysis with pre-determined inclusion and selection criteria can still have cherry picking. In addition, if still cherry-picking exist, I think the model is not simply select the top several results. No valid meta-analysis is conducted as authors described.

2. I can see it is useful to specify fixed effects models and random effects models. However, I think the assumption of cherry-picking need more supporting evidence. I do not think the assumption is correct.

3. The authors need to provide more literature on cherry-picking assumptions especially how other persons are doing that kind of assumption. I think that assumption need to be modified.

4. How to avoid or reduce cherry picking? The authors did not discuss on it.

5. To me, when I create 10000 standard normal random variables, I cherry pick the top 1000, 5000 observations, what is the distribution, mean and variance of it? It will be great if the authors can make discussion or derive this type conditional distribution based on top p% percentiles. 

6. The authors need to discuss/study other bias in addition to cherry-picking bias. To me, I think cherry-picking is only of second importance since it can be avoid by pre-specifying the rule in meta-analysis, and let the meta-analyst to conduct analysis in a standard/formal way not guided by results. Other types of bias are more important and deserves discussion and mathematical formulae and derivations. 

Round 2

Reviewer 1 Report

I originally submitted an opinion that the manuscript is worthy of publication, and the authors' responses to the minor questions I raised are sound.

I can consent to the publication of this manuscript in its current state. 

Reviewer 2 Report

All my comments have been well addressed.